# A chemical proteomic atlas of brain serine hydrolases identifies cell type-specific pathways regulating neuroinflammation

Andreu Viader[1,2]*, Daisuke Ogasawara[1,2], Christopher M Joslyn[1,2], Manuel Sanchez-Alavez[1], Simone Mori[1], William Nguyen[1], Bruno Conti[1], Benjamin F Cravatt[1,2]*

[1]The Skaggs Institute for Chemical Biology, The Scripps Research Institute, La Jolla, United States; [2]Department of Chemical Physiology, The Scripps Research Institute, La Jolla, United States

**Abstract** Metabolic specialization among major brain cell types is central to nervous system function and determined in large part by the cellular distribution of enzymes. Serine hydrolases are a diverse enzyme class that plays fundamental roles in CNS metabolism and signaling. Here, we perform an activity-based proteomic analysis of primary mouse neurons, astrocytes, and microglia to furnish a global portrait of the cellular anatomy of serine hydrolases in the brain. We uncover compelling evidence for the cellular compartmentalization of key chemical transmission pathways, including the functional segregation of endocannabinoid (eCB) biosynthetic enzymes diacylglycerol lipase-alpha (DAGL$\alpha$) and –beta (DAGL$\beta$) to neurons and microglia, respectively. Disruption of DAGL$\beta$ perturbed eCB-eicosanoid crosstalk specifically in microglia and suppressed neuroinflammatory events in vivo independently of broader effects on eCB content. Mapping the cellular distribution of metabolic enzymes thus identifies pathways for regulating specialized inflammatory responses in the brain while avoiding global alterations in CNS function.

*For correspondence: aviader@scripps.edu (AV); cravatt@scripps.edu (BFC)

## Introduction

The brain is a highly heterogeneous organ composed of diverse cell types with distinct structures, molecular compositions, and complementary functions (*Zhang et al., 2014*). Major brain cell classes include neurons, which are the principal cells responsible for directing and transmitting information in the nervous system in the form of chemical and electrical signals, and glia, which are often referred to as support cells, but are now recognized to play critical roles in regulating neurotransmission, synaptic activity, and higher-order neurophysiological, behavioral, and disease processes (*Barres, 2008*, *Gundersen et al., 2015*). Understanding the biochemical pathways that furnish neurons and glia with different properties and functions requires knowledge of the content, activity, and regulation of proteins within these cells.

Global gene expression studies have provided valuable insights into the molecular composition of different brain cell types (*Cahoy et al., 2008*, *Doyle et al., 2008*, *Beutner et al., 2013*, *Chiu et al., 2013*, *Zhang et al., 2014*). In particular, a recent study using deep RNA sequencing generated a quantitative, high-resolution map of transcript abundances for major mouse brain cell types, including neurons, astrocytes, and microglia, among others (*Zhang et al., 2014*). Thousands of cell type-enriched transcripts and splicing isoforms were identified, underscoring the unique molecular composition of different brain cell classes (*Zhang et al., 2014*). How these transcriptional signatures relate to the protein content of brain cell types, however, remains mostly unknown.

**eLife digest** The brain is made up of many types of cells. These include the neurons that transmit messages throughout the nervous system, and microglia, which act as the first line of the brain's immune defense. The activity of both neurons and microglia can be influenced by molecules called endocannabinoids that bind to proteins on the cells' surface. For example, endocannabinoids affect how a neuron responds to messages sent to it from a neighbouring neuron, and help microglia to regulate the inflammation of brain tissue.

Enzymes called serine hydrolases play important roles in several different signaling processes in the brain, including those involving endocannabinoids. Viader et al. have now studied the activities of these enzymes – including two called DAGLα and DAGLβ – in the mouse brain using a technique called activity-based protein profiling. This revealed that DAGLα plays an important role in controlling how neurons respond to endocannabinoids, while DAGLβ performs the equivalent role in microglia.

When Viader et al. shut down DAGLβ activity, this only affected endocannabinoid signaling in microglia. This also had the effect of reducing inflammation in the brain, without affecting how endocannabinoids signal in neurons. These results suggest that inhibitors of DAGLβ could offer a way to suppress inflammation in the brain, which may contribute to neuropsychiatric and neurodegenerative diseases, while preserving the normal pathways that neurons use to communicate with one another.

Several proteomic studies have been performed on isolated brain cells, in particular, neurons and astrocytes, using approaches that include two-dimensional gel electrophoresis and shotgun liquid chromatography-mass spectrometry (LC-MS) (*Yu et al., 2004*, *Yang et al., 2005*, *Glanzer et al., 2007*, *Liao et al., 2008*, *Bell-Temin et al., 2012*). Many proteins, however, are regulated in their function by post-translational mechanisms (*Kobe and Kemp, 1999*) that are difficult to discern with conventional expression-based proteomic methods. Activity-based protein profiling (ABPP) is a chemical proteomic method that uses active site-directed chemical probes to selectively target subsets of proteins in the proteome based on shared mechanistic and/or structural features (*Cravatt et al., 2008*). Because ABPP probes modify protein targets based on conserved functional features, these reagents can detect, enrich, and identify many members of a class of proteins and illuminate changes in protein activity that are not reflected in transcript or protein abundance (*Jessani et al., 2004*). ABPP has been used to map deregulated enzyme activities in a variety of physiological and pathological processes, including cancer (*Jessani et al., 2004*, *Nomura et al., 2010*, *Kohnz et al., 2015*), metabolic disorders (*Dominguez et al., 2014*), and infectious diseases (*Greenbaum et al., 2002*, *Bottcher and Sieber, 2008*, *Heal and Tate, 2012*, *Nasheri et al., 2013*).

Among the classes of proteins that have been addressed by ABPP, the serine hydrolases are a particularly large and diverse enzyme family that plays many key roles in the nervous system (*Simon and Cravatt, 2010*, *Long and Cravatt, 2011*). Serine hydrolases regulate proteolysis at the synapse to modulate neuronal plasticity (*Melchor and Strickland, 2005*), the post-translational modification state of key brain signaling proteins (*Sontag et al., 2007*, *Siegel et al., 2009*), and, perhaps most notably, the metabolism of a wide range of chemical messengers, including neurotransmitters (e.g., acetylcholine [*Phillis, 2005*], neuropeptides (e.g., α-melanocyte-stimulating hormone [*Wallingford et al., 2009*]), and lipid messengers (e.g., endocannabinoids; eCBs [*Blankman and Cravatt, 2013*]).

Here we use ABPP combined with shotgun LC-MS (*Washburn et al., 2001*) to generate an in-depth portrait of serine hydrolase activities across three major mouse brain cell types—neurons, astrocytes, and microglia. We show that the output of this functional proteomic analysis correlates well with published RNA-seq data (*Zhang et al., 2014*), although there are clear exceptions of enzymes where transcript and activity are un- and even anti-correlated. We also uncover strong evidence for the compartmentalization of functionally related serine hydrolases into distinct cell types, suggesting that their respective activities are anatomically segregated in the brain. A principal example was the selective enrichment of different sets of eCB biosynthetic and degradative enzymes in neurons versus microglia. We show that these activity patterns have functional relevance and mark

pathways that regulate immunomodulatory eCB/eicosanoid signals without globally perturbing brain eCB content.

## Results

### Serine hydrolase activity profiles of mouse brain cell types

We globally assessed serine hydrolase activities in cultured mouse neurons, astrocytes, and microglia by treating proteomes from these cells with serine hydrolase-directed fluorophosphonate (FP) probes (*Figure 1A*) (*Simon and Cravatt, 2010*). Treatment with a rhodamine-coupled FP probe (*Patricelli et al., 2001*) followed by SDS-PAGE and in-gel fluorescence scanning revealed that the brain cell types express overlapping, but distinct sets of serine hydrolase activities (*Figure 1B*). We then identified these serine hydrolases by treating brain cell proteomes with a biotin-coupled FP probe (*Liu et al., 1999*), followed by enrichment with avidin chromatography, and multidimensional LC-MS (MudPIT) analysis (*Figure 1A*) (*Washburn et al., 2001*, *Jessani et al., 2005*), which detected, in total, over 70 serine hydrolase activities across neuron, astrocyte, and microglia proteomes (*Figure 1—source data 1*). We assessed the relative amount of each serine hydrolase activity across brain cell types by the semi-quantitative method of spectral counting (*Jessani et al., 2005*, *Nahnsen et al., 2013*). We found that the serine hydrolase activity profiles measured by ABPP-MudPIT showed excellent reproducibility across the four biological replicates performed for each cell type, with high correlations for replicates from the same cell type (Pearson's correlation, mean r = 0.91 ± 0.01) and much lower correlations for serine hydrolase activity profiles across differing cell types (Pearson's correlation, mean r = 0.62 ± 0.02) (*Figure 1—figure supplement 1A*). Microglia and astrocytes displayed the highest pairwise correlation across the brain cell types examined, suggesting greater similarity of serine hydrolase metabolic activities among glia than between neurons and glia (*Figure 1—figure supplement 1A*).

Unsupervised hierarchical clustering of the ABPP-MudPIT data highlighted distinct groups of serine hydrolases that were enriched in specific brain cell types (*Figure 1C*; *Figure 1—figure supplement 1B*). These enrichment profiles included some enzymes with established neuronal expression, such as acetylcholine esterase (ACHE) (*Hicks et al., 2011*) and fatty acid amide hydrolase (FAAH) (*Egertova et al., 1998*, *Tsou et al., 1998*), which catalyze the hydrolysis of the neurotransmitter acetylcholine and eCB anandamide, respectively (*Figure 1C*; *Figure 1—figure supplement 1B*), as well as many less well characterized enzymes with strong preferential activity in astrocytes (e.g., ABHD2, ABHD16A) and microglia (e.g. DAGLβ, AOAH) (*Figure 1C*; *Figure 1—figure supplement 1B*). Brain cell type-enriched serine hydrolases spanned a wide range of spectral count values (*Figure 1—figure supplement 1C*), indicating that restricted cellular distribution was not correlated with the abundance of detected enzyme activities.

Comparison of our ABPP data with recently reported mRNA expression results generated by RNA-seq (*Zhang et al., 2014*) revealed a good overall correlation across mouse brain cell types (Pearson's correlation r = 0.33, ****p < 0.0001), particularly for those serine hydrolases displaying prominent cell type-specific enrichment (Pearson's correlation r = 0.54, **p < 0.01; *Figure 1D*). We also observed, however, a few notable exceptions of enzymes where transcript and activity were poorly correlated, including enzymes that showed i) near-equivalent RNA-seq signals across all three brain cell types, but predominant activity signals in one cell type (e.g., ABHD16A); ii) strong RNA-seq and activity signals in distinct cell types (e.g., HTRA1), and iii) strong RNA-seq and activity signals that were anti-correlated across all three cell types (e.g. MGLL) (*Figure 1E*).

These data, taken together, demonstrate that ABPP furnishes an in-depth portrait of serine hydrolase activities across neurons, astrocytes, and microglia that complements and bolsters the information afforded by expression-based analytical platforms (e.g. RNA-seq) and illuminates candidate enzymes that contribute to the metabolic specialization of major brain cell types.

### Microglia express a distinct complement of 2-AG metabolic enzymes

Prominent among serine hydrolases displaying strong brain cell-enriched profiles were multiple biosynthetic and degradative enzymes of the eCB 2-arachidonoyl glycerol (2-AG) (*Mechoulam et al., 1995*, *Sugiura et al., 1995*). This arachidonic acid (AA)-derived retrograde lipid messenger broadly modulates synaptic function, neurophysiology, and behavior (*Fowler et al., 2005*, *Pacher et al.,*

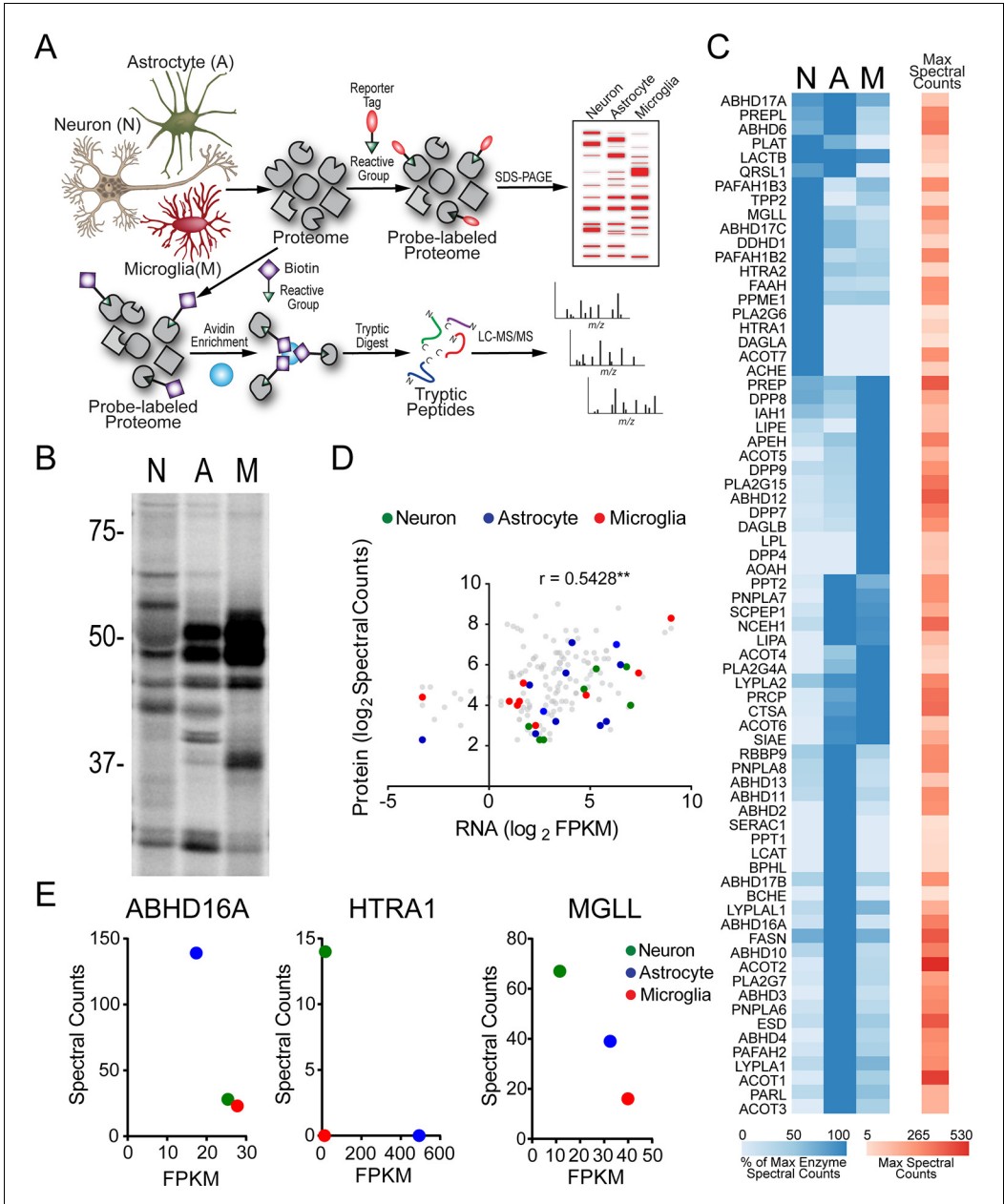

**Figure 1.** Serine hydrolase activity profiles of mouse brain cell types. (A) Cartoon scheme of gel- and MS-based activity-based protein profiling (ABPP) methods used to measure serine hydrolase activities in primary mouse neurons, astrocytes, and microglia. For gel-based ABPP, a fluorophosphonate (FP) reactive group conjugated to a rhodamine reporter tag (red oval; FP-Rh) is used (*Patricelli et al., 2001*). For MS-based ABPP (ABPP-MudPIT), an FP reactive group conjugated to a biotin reporter tag (purple diamond; FP-biotin) is used (*Liu et al., 1999*). (B) Gel-based ABPP of membrane proteomes from different brain cell types. (C) Hierarchically clustered heatmap of ABPP-MudPIT data (left) for serine hydrolases detected in neurons (N), astrocytes (A) and microglia (M). Data represent the mean spectral count values for each serine hydrolase (from four independent experiments) expressed as % of cell type with maximum number of spectral counts (right heatmap shows the maximum spectral counts among cell types for each serine hydrolase). (D) Relationship between serine hydrolase activities, as measured by ABPP-MudPIT, and previously reported mRNA expression for these enzymes, as measured by RNA-Seq (*Zhang et al., 2014*), in neurons, astrocytes, and microglia. Serine hydrolases showing $\geq$ three-fold enrichment in activity in a specific cell type as measured by ABPP-MudPIT are shown as filled colored circles and a Pearson's correlation reported for the aggregate correlation between their ABPP and RNAseq profiles (r = 0.54; p < 0.01). (E) Examples of serine hydrolases where activity and mRNA expression measurements were uncorrelated (ABHD16A, HTRA1) or anti-correlated (MGLL).

The following source data and figure supplements are available for figure 1:

**Source data 1.** Serine hydrolases identified in neuron, astrocyte, and microglia proteomes by ABPP-MudPIT.

*Figure 1 continued on next page*

*Figure 1 continued*

**Figure supplement 1.** Serine hydrolase activity profiles of mouse brain cell types.

**Figure supplement 2.** DAGLβ and ABHD12 activities are enriched in microglia.

*2006*, *Kano et al., 2009*) by binding to G protein-coupled cannabinoid receptors $CB_1$ ($CB_1R$) and $CB_2$ ($CB_2R$), which are also targets of the primary psychoactive component of marijuana $\Delta^9$-tetrahydrocannabinol (*Mechoulam and Hanus, 2000*). Enzymatic hydrolysis of 2-AG also serves as a major source of AA for the synthesis of brain eicosanoids that regulate neuroinflammatory processes (*Nomura et al., 2011*, *Chen et al., 2012*, *Piro et al., 2012*). Previous studies have demonstrated that diacylglycerol lipase-α (DAGLα) (*Bisogno et al., 2003*) and monoacylglycerol lipase (MGLL) (*Karlsson et al., 1997*) are the principal 2-AG biosynthetic and degradative enzymes, respectively, in the brain (*Long et al., 2009*, *Chanda et al., 2010*, *Gao et al., 2010*, *Schlosburg et al., 2010*, *Tanimura et al., 2010*). Our ABPP data revealed that both of these enzymes are preferentially expressed in neurons compared to astrocytes or microglia (*Figure 1C*). On the other hand, microglia expressed high levels of the alternative 2-AG biosynthetic and degradative enzymes DAGLβ (*Bisogno et al., 2003*) and ABHD12 (*Blankman et al., 2007*), respectively (*Figures 1C*). Previous RNA-seq measurements also supported the enriched expression of DAGLβ and ABHD12 in microglia compared to neurons or astrocytes (*Zhang et al., 2014*). We verified strong DAGLβ activity in microglia using the tailored activity-based probe HT-01 (*Hsu et al., 2012*) (*Figure 1—figure supplement 2A*) and confirmed that ABHD12 substantially contributed to 2-AG hydrolysis in microglia (and astrocytes), but not neurons using cells isolated from *Abhd12⁻/⁻* mice (*Blankman et al., 2013*) (*Figure 1—figure supplement 2B*).

These data pointed to a potential anatomical demarcation of enzyme function within the eCB system where individual brain cell types would use distinct sets of enzymes to control 2-AG metabolism and signaling, including crosstalk with other lipid networks. We next set out to test this premise by evaluating the contributions of ABHD12 and DAGLβ to regulating 2-AG metabolism in microglia.

## ABHD12 functions as a 2-AG hydrolase in microglia

ABHD12 exhibits 2-AG hydrolase activity in vitro (*Blankman et al., 2007*, *Navia-Paldanius et al., 2012*), but *Abhd12⁻/⁻* mice have normal brain 2-AG content (*Blankman et al., 2013*), and therefore the potential physiological relevance of ABHD12 as a regulator of 2-AG metabolism remains unclear. Having found that microglia from *Abhd12⁻/⁻* mice show ~40% reductions in 2-AG hydrolytic activity (*Figure 1—figure supplement 2B*), we next examined 2-AG content in these cells. *Abhd12⁻/⁻* microglia displayed a modest, but significant increase in cellular 2-AG compared to *Abhd12⁺/⁺* microglia (*Figure 2A*); in contrast, 2-AG content in *Abhd12⁻/⁻* neurons and astrocytes was unaltered and instead elevated in *Mgll⁻/⁻* neurons and astrocytes (*Figure 2A*). *Mgll⁻/⁻* microglia also showed a strong elevation in cellular 2-AG that was much greater in magnitude compared to that observed in *Abhd12⁻/⁻* microglia (*Figure 2A*). Notably, however, secreted 2-AG content was much higher from *Abhd12⁻/⁻* microglia and unaltered from *Mgll⁻/⁻* microglia compared to wild type microglia (*Figure 2B*). These data suggested that ABHD12 primarily regulates extracellular pools of 2-AG in microglia (*Figure 2B*), which is consistent with the enzyme's luminal/extracellular orientation (*Blankman et al., 2007*). Secreted 2-AG was unaltered in *Abhd12⁻/⁻* neurons and astrocytes (but elevated in *Mgll⁻/⁻* neurons) (*Figure 2B*), indicating that ABHD12 specifically regulates extracellular 2-AG content in microglia. Consistent with this extracellular pool of 2-AG being competent for signaling, we observed tonically increased CBR-dependent ERK1/2 phosphorylation in *Abhd12⁻/⁻* microglia (*Figure 2C and D*).

We next evaluated whether ABHD12-mediated 2-AG hydrolysis provided an AA source for prostaglandin production in microglia. We found that *Abhd12⁺/⁺* and *Abhd12⁻/⁻* microglia exhibited similar concentrations of prostaglandins, either basally or after stimulation with bacteria-derived lipopolysaccharide (LPS; 100 ng/mL, 4 hr) (*Figure 2E*) and did not show differences in LPS-induced cytokine production (*Figure 2F*). In contrast, treatment of microglia with the MGLL inhibitor KML29 (*Chang et al., 2012*) significantly lowered basal and LPS-induced prostaglandins (*Figure 2G*), as well

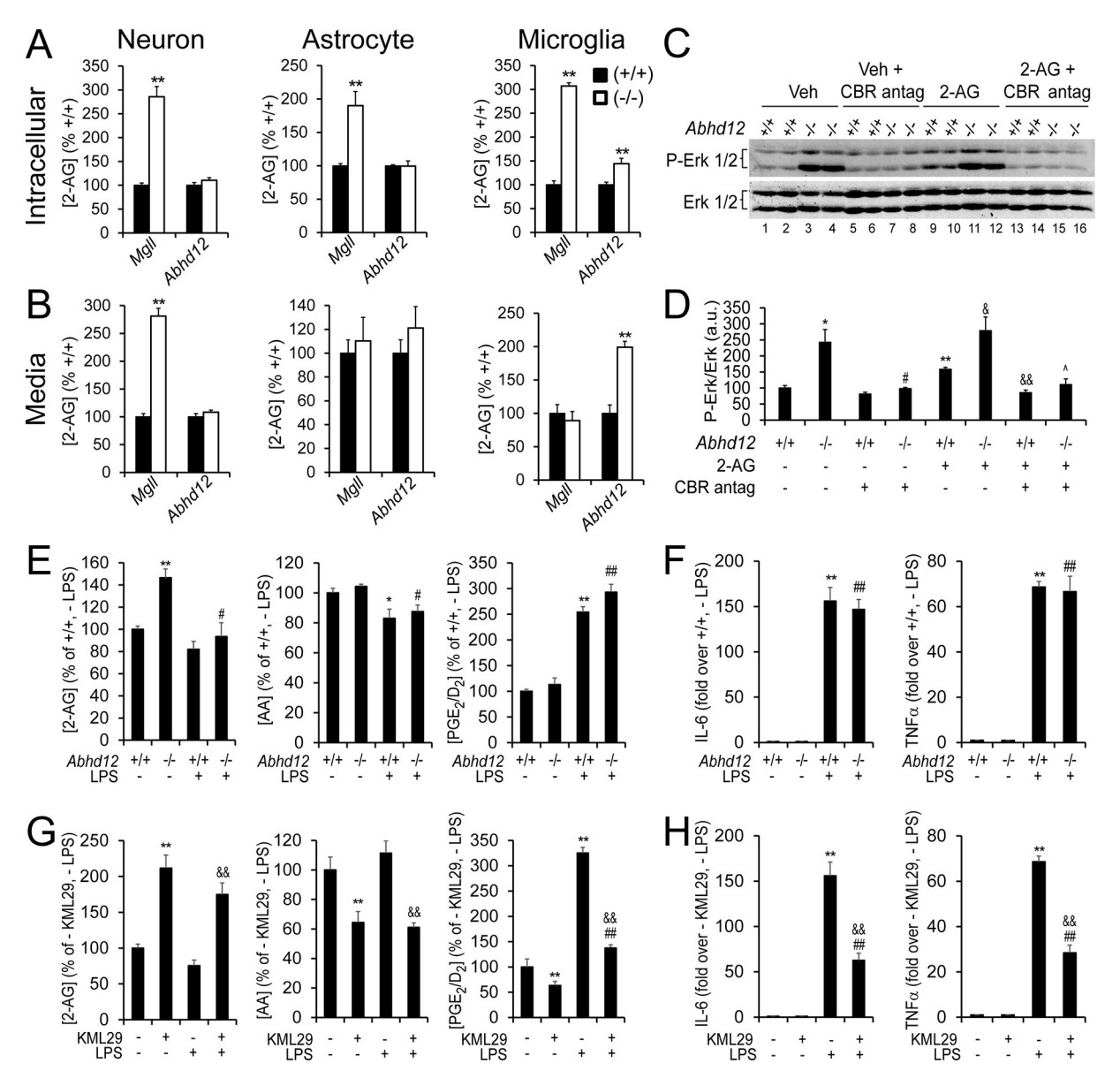

**Figure 2.** ABHD12 modulates 2-AG content and CBR activation, but not eCB-eicosanoid crosstalk in microglia. (A,B) Intracellular (A) and secreted (B) 2-AG content in cultured wild type, *Abhd12*−/− and *Mgll*−/− neurons, astrocytes, and microglia. Data represent average values ± SEM; N = 4-6 per genotype. **p < 0.01 for *Abhd12*−/− or *Mgll*−/− cells vs corresponding wild-type cells. (C,D) Immunoblot (C) and quantification of band density (D) for Erk1/2 phosphorylation, which increases upon 2-AG-mediated activation of CBRs (*Walter et al., 2003*), in wild-type and *Abhd12*−/− microglia. *Abhd12*−/− microglia display hyperphosphorylation of Erk1/2 (lanes 1-4), which can be partially mimicked in *Abhd12*+/+ cells by addition of exogenous 2-AG (1 μM, 15 min; lanes 9–12) and fully blocked by pre-treatment with CBR1 and CBR2 antagonists (rimonabant and AM630, respectively) (5 μM each, 1 hr pre-treatment; lanes 5-8 and 13-16). N = 4 per genotype and treatment. Immunoblot shows two representative replicate experiments for each group. Data represent average values ± SEM; N = 4 per genotype and treatment; *p < 0.05 and **p < 0.01 for vehicle-treated *Abhd12*−/− group and 2-AG-treated *Abhd12*+/+ group vs vehicle-treated *Abhd12*+/+ group; #p < 0.05 for CBR antagonist-treated *Abhd12*−/− group vs vehicle-treated *Abhd12*−/− group; &p < 0.05 and &&p < 0.01 for 2-AG-treated *Abhd12*−/− or 2-AG/CBR antagonist-treated *Abhd12*+/+ groups vs 2-AG-treated *Abhd12*+/+ group; ^p < 0.05 for 2-AG/CBR antagonist-treated *Abhd12*−/− group vs 2-AG-treated *Abhd12*−/− group. (E,G) 2-AG, AA, and PGE2/D2 content basally and following exposure to the pro-inflammatory agent LPS (100 ng/mL for 4 hr) in *Abhd12*−/− and *Abhd12*+/+ microglia (E) or in wild-type microglia pre-treated with the MGLL inhibitor KML29 (250 nM, 3 hr prior to LPS; G). Data represent average values ± SEM; N = 5 per genotype and treatment. *p < 0.05 and **p < 0.01 for vehicle-treated *Abhd12*−/− microglia and LPS-treated *Abhd12*+/+ microglia groups (E) or KML29-treated wild-type microglia and LPS-treated wild type microglia groups (G) vs vehicle-treated *Abhd12*+/+ microglia or vehicle-treated wild-type microglia groups, respectively; #p < 0.05 and ##p < 0.01 for LPS-treated *Abhd12*−/− microglia (E) or KML29-treated, LPS-treated wild-type microglia (G) groups vs vehicle-treated *Abhd12*+/+ microglia or KML29-treated wild-type microglia groups, respectively; &&p < 0.01 for KML29-treated, LPS-treated wild-type microglia group vs LPS-treated wild-type

*Figure 2 continued on next page*

*Figure 2 continued*

microglia group. (**F,H**) Cytokine production basally and following exposure to LPS (100 ng/mL for 4 hr) as measured by ELISA in *Abhd12*$^{-/-}$ and *Abhd12*$^{+/+}$ microglia (F) or in wild-type microglia pre-treated with KML29 (250 nM, 3 hr prior to LPS; H). Data represent average values ± SEM; N = 5 per genotype and treatment; \*\*p < 0.01 for LPS-treated *Abhd12*$^{+/+}$ microglia (F) or LPS-treated wild-type microglia (H) groups vs vehicle-treated *Abhd12*$^{+/+}$ microglia or vehicle-treated wild-type microglia groups, respectively; $^{\#\#}$p < 0.01 for LPS-treated *Abhd12*$^{-/-}$ microglia (F) or KML29-treated, LPS-treated wild-type microglia (G) groups vs vehicle-treated *Abhd12*$^{-/-}$ or KML29-treated wild-type microglia groups, respectively; $^{\&\&}$p < 0.01 for KML29-treated, LPS-treated wild-type microglia group vs LPS-treated wild-type microglia group.

as LPS-induced cytokine production (*Figure 2H*). These data are consistent with recent reports showing that MGLL-inactivated microglia are impaired in LPS-induced prostaglandin production and inflammatory responses (*Pihlaja et al., 2015*, *Viader et al., 2015*).

Our results thus indicate that ABHD12 is a major regulator of secreted 2-AG and CBR activation in microglia, while MGLL is the primary intracellular 2-AG hydrolase in these cells. Considering that the prostaglandin biosynthetic enzymes PTGS1 and PTGS2 are also localized intracellularly (*Soberman and Christmas, 2003*), it is perhaps not surprising that eCB-eicosanoid crosstalk was also found to be governed by MGLL rather than ABHD12 in microglia.

## DAGLβ is a principal 2-AG biosynthetic enzyme in microglia

We next examined the respective roles of DAGLα and DAGLβ in modulating eCB signaling pathways in brain cell types. Neurons and astrocytes from *Dagla*$^{-/-}$ mice displayed much lower 2-AG content compared to neurons and astrocytes from wild type or *Daglb*$^{-/-}$ mice, indicating that DAGLα is responsible for the bulk of 2-AG biosynthesis in these brain cell types (*Figure 3A and B*). In contrast, microglia from *Daglb*$^{-/-}$, but not *Dagla*$^{-/-}$ mice showed substantial reductions in 2-AG (*Figure 3C*). A similar profile was observed for AA, which was lower in neurons and astrocytes from *Dagla*$^{-/-}$ mice and microglia from *Daglb*$^{-/-}$ mice (*Figure 3A–C*), as well as for prostaglandins, with the exception that we did not detect prostaglandins in neurons (*Figure 3A–C*). We also observed comparable lipid changes in wild-type microglia treated with two DAGL inhibitors (KT109 and KT172), but not with a structurally related inactive control compound (KT195) (*Hsu et al., 2012*) (*Figure 3—figure supplement 1*).

We next assessed the lipid profiles of *Daglb*$^{+/+}$ and $^{-/-}$ microglia following LPS stimulation (100 ng/mL, for 4 hr) and found that DAGLβ inactivation resulted in the coordinated decreases in intracellular 2-AG, AA, and prostaglandins (*Figure 4A–C*). No changes in these lipids were observed in *Dagla*$^{-/-}$ microglia. *Daglb*$^{-/-}$ microglia, but not *Dagla*$^{-/-}$ microglia also displayed marked reductions in LPS-induced inflammatory cytokine production compared to wild type microglia (*Figure 4D–J*). Similar lipid and cytokine reductions were observed in LPS-stimulated wild type microglia that were pre-treated (3 hr) with KT172, but not KT195 (*Figure 4K–U*).

These results, taken together, demonstrate that distinct DAGL enzymes are responsible for controlling eCB and eicosanoid metabolism in astrocytes and neurons (DAGLα) versus microglia (DAGLβ), and that blocking DAGLβ genetically or pharmacologically attenuates LPS-induced inflammatory responses in microglia.

## Effects of disrupting DAGLβ on brain lipid metabolism and inflammatory responses in vivo

*Dagla*$^{-/-}$ mice have been shown to display dramatic reductions in brain 2-AG (~80-90%), demonstrating that DAGLα is responsible for the bulk of 2-AG biosynthesis in the CNS (*Gao et al., 2010*, *Tanimura et al., 2010*). The contribution of DAGLβ is less clear, as *Daglb*$^{-/-}$ mice have been reported to show either no changes (*Hsu et al., 2012*, *Tanimura et al., 2012*) or modest decreases (*Gao et al., 2010*) in brain 2-AG. Our data generally matched these previous studies, as we observed severely depleted brain 2-AG (~90%, *Figure 5A*) in *Dagla*$^{-/-}$ mice, which was accompanied by a substantial increase in 1-stearoyl-2-arachidonoyl-sn-glycerol (SAG) (*Figure 5B*), a main precursor of 2-AG (*Shonesy et al., 2014*, *Ogasawara et al., 2015*), and corresponding reductions in brain AA (*Figure 5C*) and prostaglandins (*Figure 5D*). In contrast, brain 2-AG, SAG, and AA content were unaltered in *Daglb*$^{-/-}$ mice (*Figure 5A–C*), although these animals did exhibit a significant reduction in brain PGE$_2$ (*Figure 5D*), perhaps owing to the modulation of 2-AG and related lipids selectively in

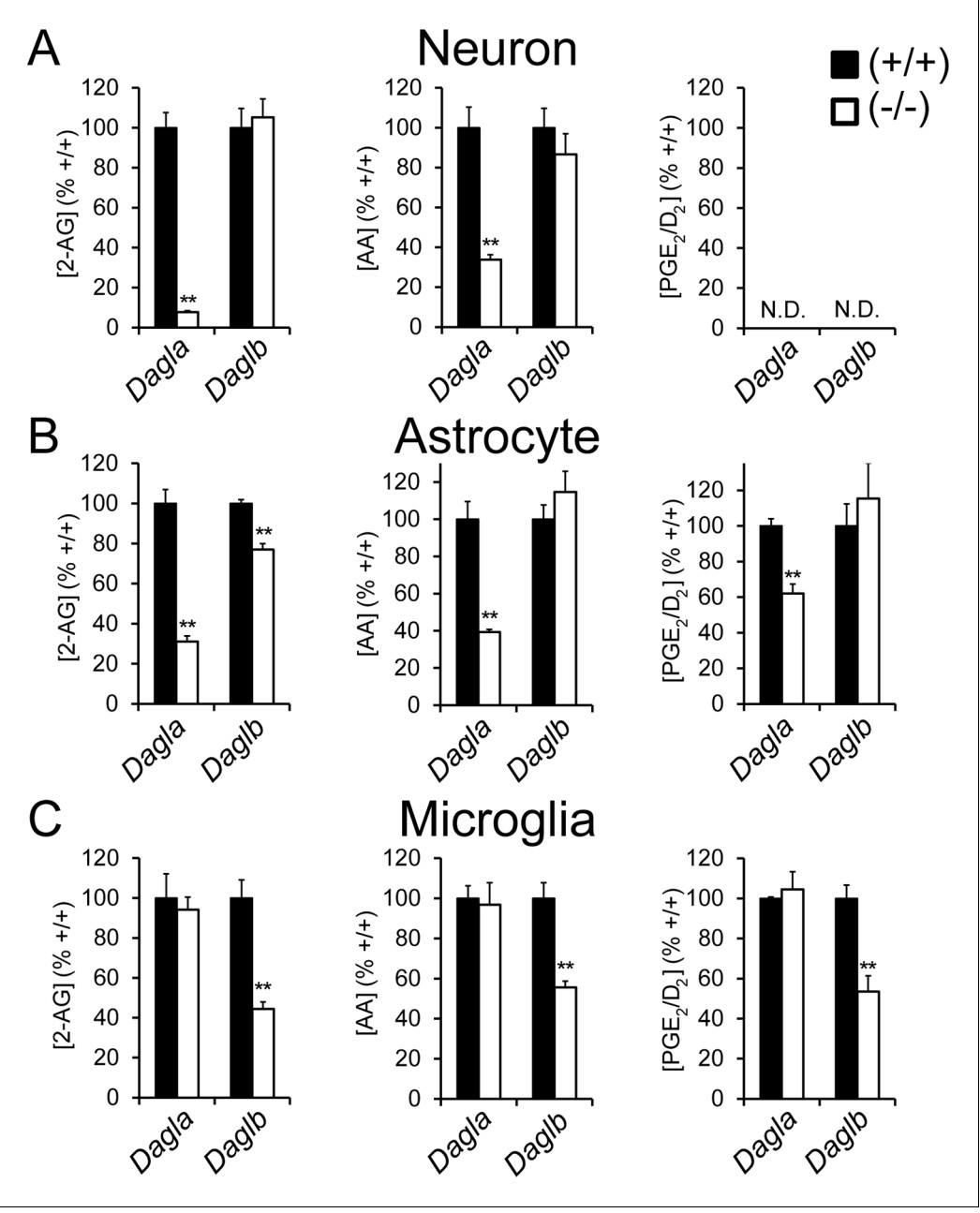

**Figure 3.** DAGLβ is a principal 2-AG biosynthetic enzyme in microglia. (A–C) 2-AG, AA, and PGE$_2$/D$_2$ content in primary neurons (A), astrocytes (B), and microglia (C) derived from $Dagla^{-/-}$, $Daglb^{-/-}$ and corresponding wild-type mice. Data represent average values ± SEM; N = 5–8 per genotype. **p < 0.01 for $Dagla^{-/-}$ or $Daglb^{-/-}$ groups vs corresponding wild-type groups.

The following figure supplement is available for figure 3:

**Figure supplement 1.** Pharmacological blockade of DAGLβ modulates eCB/eicosanoid metabolism in microglia.

microglia, which, despite representing only 10% of all brain cells (**Aguzzi et al., 2013**), display enriched expression of the prostaglandin synthase PTGS1 (or cyclooxygenase-1 (COX-1)) (**Hoozemans et al., 2001**, **Font-Nieves et al., 2012**, **Zhang et al., 2014**).

To more directly address whether a DAGLβ-dependent pool of 2-AG exists in the brain, we evaluated the inhibition of DAGLβ in the context of $Dagla^{-/-}$ mice. We first found that the

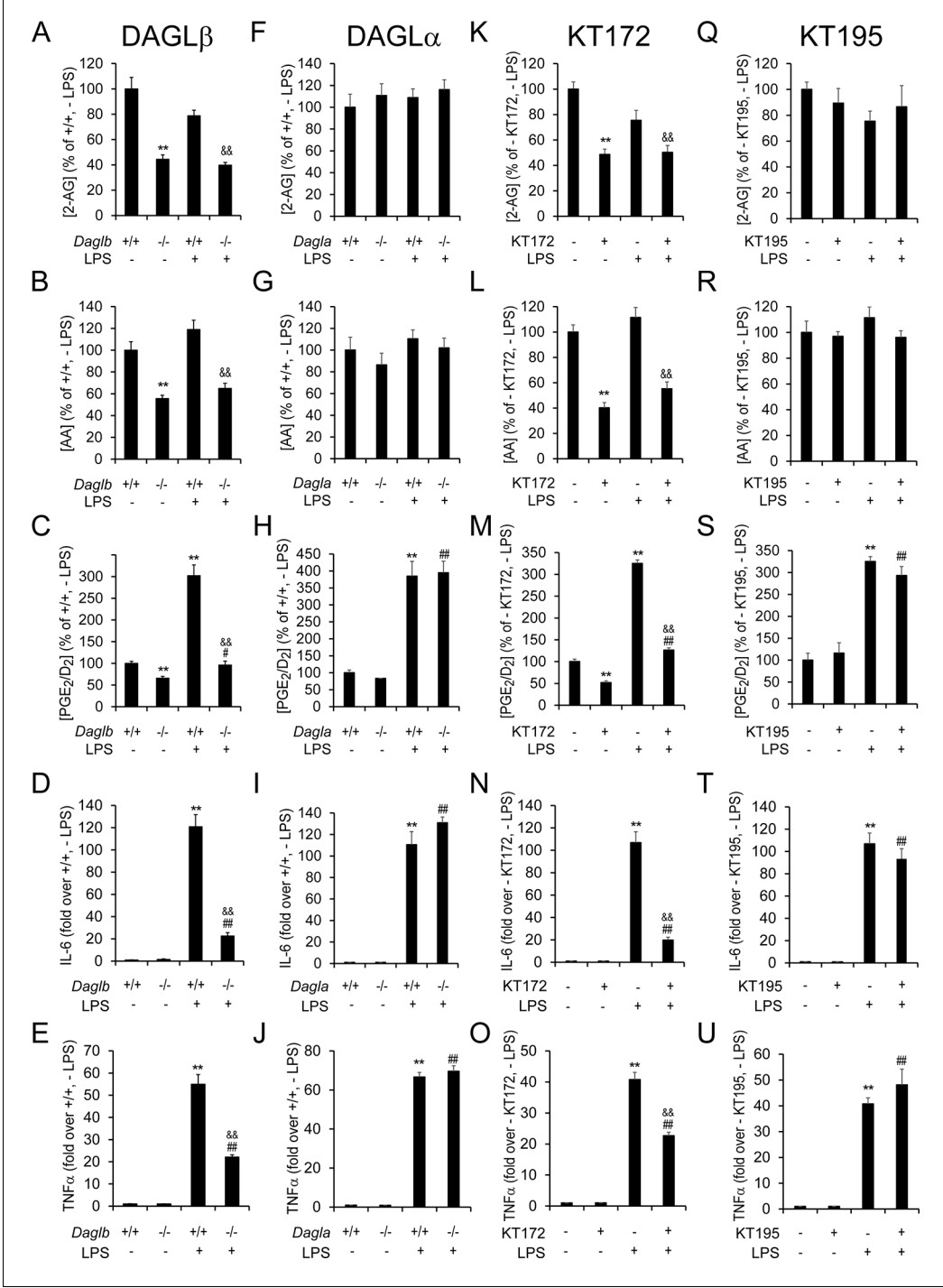

**Figure 4.** Genetic or pharmacologic inactivation of DAGLβ impairs LPS-induced eCB-eicosanoid crosstalk and cytokine production in microglia. (A–C, F–H, K–M, Q–S) 2-AG, AA, and PGE$_2$/D$_2$ content basally and following exposure to LPS (100 ng/mL for 4 hr) in *Daglb$^{-/-}$* (A-C), *Dagla$^{-/-}$* (F–H) and corresponding wild-type microglia, or in wild-type microglia treated with the DAGL inhibitor-treated KT172 (500 nM, 3 hr prior to LPS; K–M) or an inactive control compound (KT195, 500 nM, 3 hr prior to LPS; Q–S). Data represent average values ± SEM; N = 5 per genotype and treatment. (D,E,I,J,N,O,T,U) Cytokine production basally and following exposure to LPS (100 ng/mL for 4 hr) as measured by ELISA in *Daglb$^{-/-}$* (D,E), *Dagla$^{-/-}$* (I,J) and corresponding wild-type microglia, or in wild-type microglia treated with KT172 (500 nM, 3 hr prior to LPS; N, O) or a KT195 (500 nM, 3 hr prior to LPS; T, U). Data represent average values ± SEM; N = 5 per genotype and treatment. For A–U, **p < 0.01 for *Daglb$^{-/-}$* or KT172-treated wild-type microglia or LPS-treated *Daglb$^{+/+}$*, *Dagla$^{+/+}$*, and wild-type microglia groups vs

*Figure 4 continued on next page*

*Figure 4 continued*

corresponding vehicle-treated wild-type microglia groups; #p < 0.05 and ##p < 0.01 for LPS-treated *Dagl*−/− or inhibitor (KT172 or KT195)-treated groups vs corresponding vehicle-treated *Dagl*−/− or inhibitor (KT172 or KT195)-treated groups; &&p < 0.01 for LPS-treated *Daglb*−/− or LPS-treated, KT172-treated groups vs corresponding LPS-treated *Daglb*+/+ or LPS-treated, vehicle-treated groups.

pharmacological blockade of 2-AG hydrolysis with the MGLL inhibitor MJN110 (*Niphakis et al., 2013*) promoted a significant recovery of brain 2-AG content in *Dagla*−/− mice (*Figure 5E and F*), consistent with a previous report (*Shonesy et al., 2014*). We then tested inhibition of DAGLβ using a recently described CNS-active DAGL inhibitor DO34 (*Ogasawara et al., 2015*) and found that a high dose of this compound (100 mg/kg, i.p.) near-completely blocked DAGLβ activity in brain (*Figure 5E*) and suppressed the MJN110-induced rescue of brain 2-AG in *Dagla*−/− mice (*Figure 5F*). These data thus indicate that DAGLβ can contribute to bulk 2-AG content in the CNS, in particular, in the absence of DAGLα.

MGLL blockade has been shown to lower prostaglandins and attenuate neuroinflammation in the brain (*Nomura et al., 2011*, *Chen et al., 2012*, *Piro et al., 2012*). These previous findings, combined with the dramatic reductions in brain 2-AG, AA, and prostaglandins observed in *Dagla*−/−mice (*Figure 5A–D*) and the substantial suppression of inflammatory responses in *Daglb*−/−microglia (*Figure 4*), motivated us to test whether microglial activation, a common feature of diverse nervous system diseases (*Aguzzi et al., 2013*), was regulated by DAGL enzymes in vivo.

Wild type and DAGL-disrupted mice were treated with LPS (1 mg/kg, once per day for four days) and then sacrificed and their lipid profiles and microglial activation state analyzed. This LPS treatment paradigm resulted in a significant increase in brain PGE$_2$, but other measured lipids (2-AG, SAG, AA, PGD$_2$) were unaltered. As anticipated, *Dagla*−/− mice showed substantial reductions in brain 2-AG, AA, and prostaglandins, along with elevations in SAG, in both control and LPS-treated mice (*Figure 6A*). Along with these lipids changes, *Dagla*−/− mice also displayed attenuated LPS-induced microglial activation in the brain (*Figure 6B* and *Figure 6—figure supplement 1*). In contrast, *Daglb*−/− mice challenged with LPS did not show changes in the measured brain lipids, including PGE2, for which an LPS-stimulated increase appeared to override the basal reduction observed in *Daglb*−/− mice (*Figure 6C*). Yet, LPS-treated *Daglb*−/− mice still exhibited reduced activation of brain microglia (*Figure 6D* and *Figure 6—figure supplement 1*). No obvious differences in basal cell morphology of microglia were observed between wild-type and *Dagla*−/− or *Daglb*−/− mice (*Figure 6 B,D*, and *Figure 6—figure supplement 1*). LPS-induced anapyrexia, a profound reduction in core body temperature mediated largely by central inflammatory processes (*Romanovsky, 2004*), which we have found to be sensitive to DAGLa disruption (*Ogasawara et al., 2015*), was also greatly attenuated in *Daglb*−/− mice (*Figure 6E*). These results indicate that disruption of either DAGLα or DAGLβ curbs neuroinflammatory responses in the brain, with blockade of the latter enzyme doing so while avoiding bulk changes in eCBs and eicosanoids.

## Discussion

Understanding the metabolic and signaling pathways expressed by major brain cell types should help to illuminate key features of nervous system physiology and disease (*Kreft et al., 2012*, *Kettenmann et al., 2013*, *Morrison et al., 2013*). Gene expression profiling (*Cahoy et al., 2008*, *Doyle et al., 2008*, *Beutner et al., 2013*, *Chiu et al., 2013*, *Zhang et al., 2014*) and proteomic studies (*Yu et al., 2004*, *Yang et al., 2005*, *Glanzer et al., 2007*, *Liao et al., 2008*, *Bell-Temin et al., 2012*, *Sharma et al., 2015*) performed on isolated brain cells have provided valuable insights into the unique molecular makeup of different brain cell types, but the assessment of the functional state of proteins, and not just their abundance, is also needed to dissect the cellular distribution of signaling and metabolic pathways that regulate nervous system (patho)physiology. We have herein used ABPP to provide a global portrait of serine hydrolase activities in neurons, astrocytes, and microglia, and, through doing so, identified cell type-specific regulation of key chemical transmission pathways, in particular eCBs and eicosanoids, that exert differential impact on global versus inflammation-induced processes in the mouse brain.

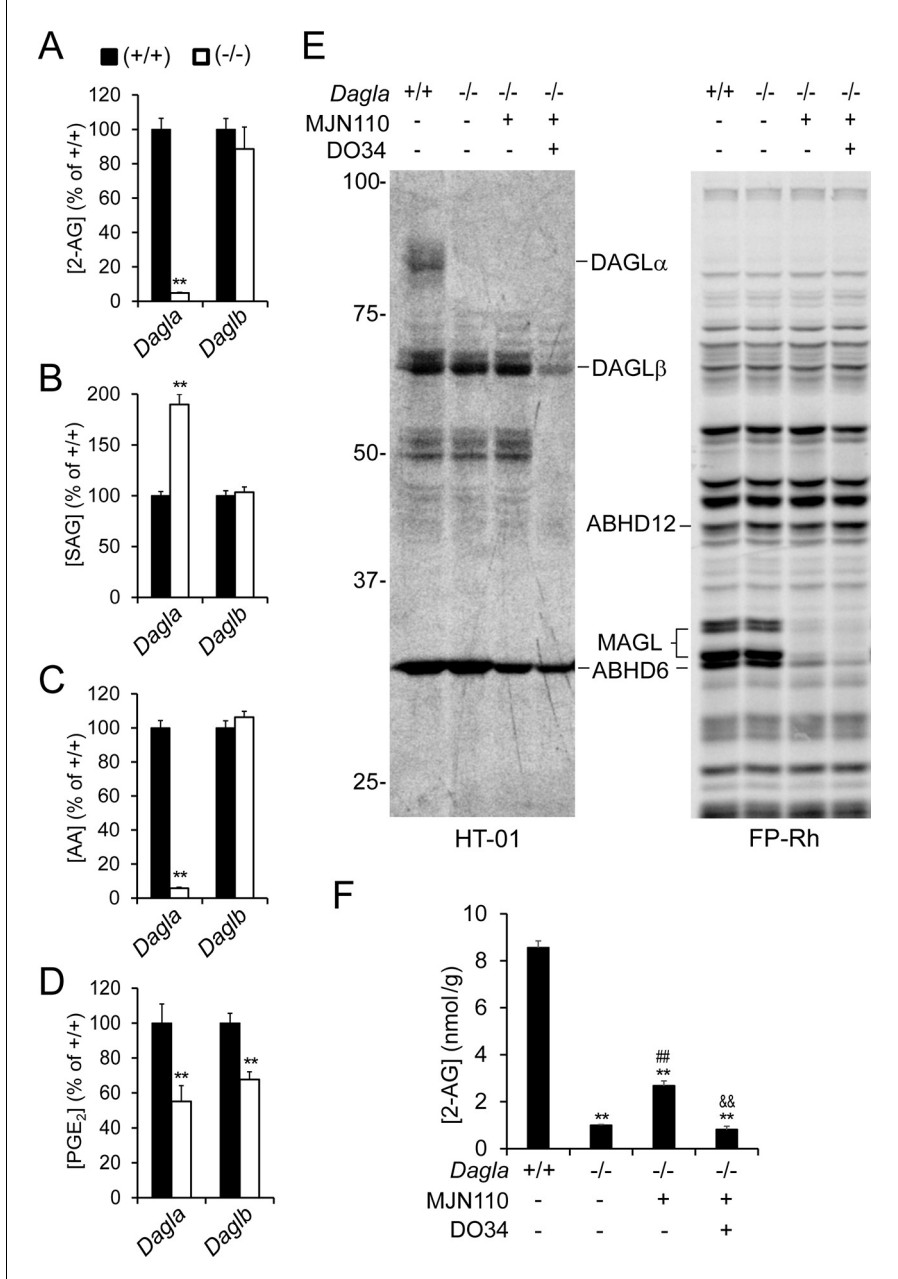

**Figure 5.** DAGLβ modulates discrete brain 2-AG pools in vivo. (**A–D**) 2-AG (A), SAG (B), AA (C), and PGE₂ (D) content in brain tissue from *Dagla⁻/⁻*, *Daglb⁻/⁻* mice and their corresponding wild-type littermates. Data represent average values ± SEM; N = 6-8 mice per genotype. **p < 0.01 for *Dagla⁻/⁻* or *Daglb⁻/⁻* mice vs corresponding wild-type mice. (**E**) Gel-based ABPP analysis of brain membrane proteomes from *Dagla⁻/⁻* mice following treatment with the MGLL inhibitor MJN110 (i.p. 10 mg/kg, 3 hr) the DAGL inhibitor DO34 (i.p. 100 mg/kg, 2 hr), or sequentially with both inhibitors (DO34, 2 hr; followed by MJN110, 3 hr) using a DAGL-directed probe (HT-01, left) (*Hsu et al., 2012*) or the broad-spectrum serine hydrolase probe FP-Rh (E, right) (*Patricelli et al., 2001*). Fluorescent gels are shown in grayscale and 2-AG metabolic enzymes are labeled. (**F**) Brain 2-AG content from *Dagla⁻/⁻* mice following treatment with the MGLL inhibitor MJN110 (i.p. 10 mg/kg, 3 hr) or sequentially with the DAGL inhibitor DO34 (i.p. 100 mg/kg, 2 hr) and MJN110 (DO34, 2 hr; followed by MJN110, 3 hr). Data represent average values ± SEM; N = 5 per genotype and treatment. **p<0.01 for *Dagla⁻/⁻* groups vs vehicle-treated *Dagla⁺/⁺* group; ##p < 0.01 for MJN110-treated for *Dagla⁻/⁻* group vs vehicle-treated *Dagla⁻/⁻* group; &&p < 0.01 for DO34 + MJN110-treated *Dagla⁻/⁻* group vs MJN100-treated *Dagla⁻/⁻* group.

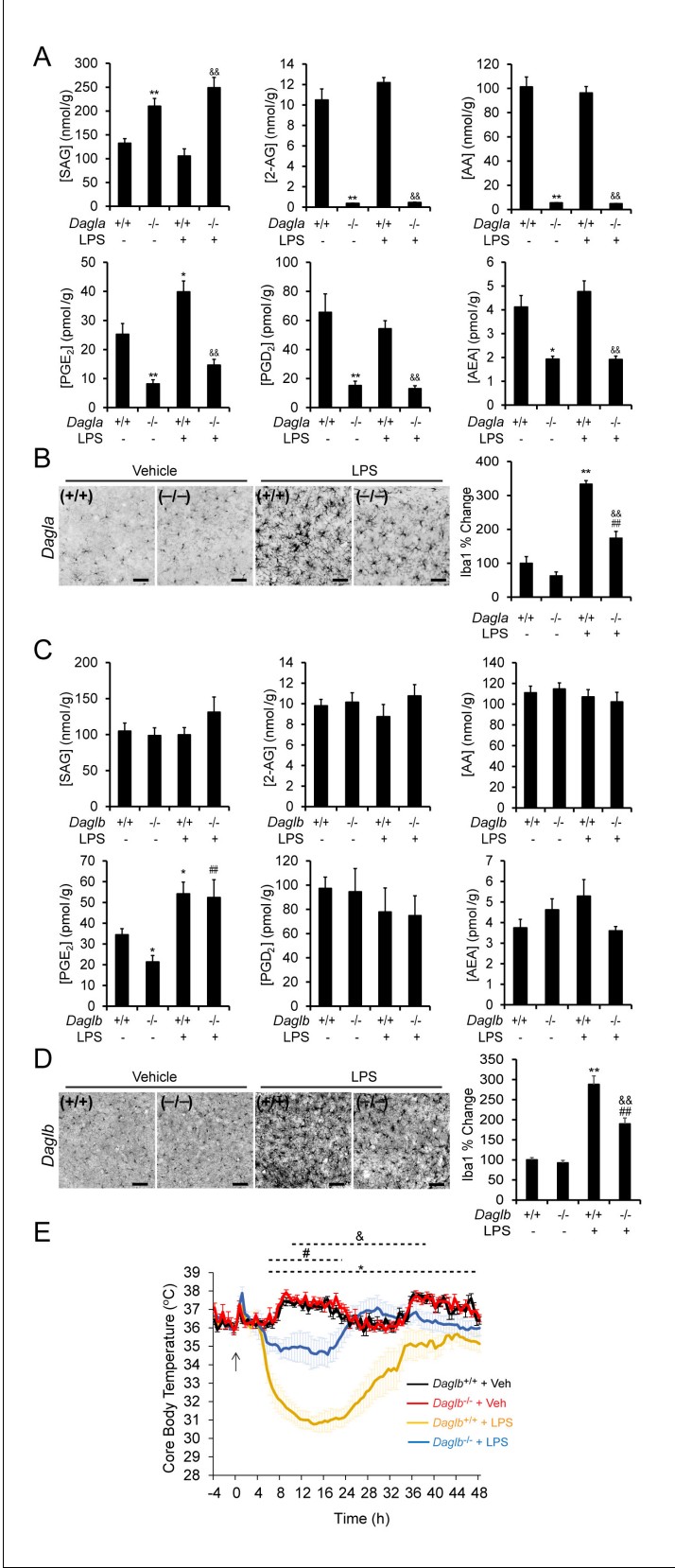

**Figure 6.** DAGL blockade attenuates microglial neuroinflammatory responses in vivo. (**A**) SAG, 2-AG, AA, PGE$_2$, PGD$_2$ and AEA content in brain tissue from *Dagla*$^{-/-}$ mice and wild-type littermates basally and following exposure
*Figure 6 continued on next page*

*Figure 6 continued*

to LPS (i.p., 1 mg/kg once per day for 4 days). Data represent average values ± SEM; N = 5-6 per genotype and treatment. *p < 0.05 and **p < 0.01 for vehicle-treated $Dagla^{-/-}$ group or LPS-treated $Dagla^{+/+}$ group vs vehicle-treated $Dagla^{+/+}$ group; $^{\&\&}$p < 0.01 LPS-treated $Dagla^{-/-}$ group vs LPS-treated $Dagla^{+/+}$ group. (B) Representative pictures and quantification of microglial activation assessed by Iba-1 staining (a microglia marker that becomes upregulated during inflammatory activation of these cells) in hippocampal regions from $Dagla^{-/-}$ mice and wild-type littermates basally and following exposure to LPS (i.p., 1 mg/kg once per day for 4 days). Scale bar, 50 µm. Data represent average values ± SEM; N = 6 per genotype and treatment. **p < 0.01 for LPS-treated $Dagla^{+/+}$ group vs vehicle-treated $Dagla^{+/+}$ group; $^{\#\#}$p < 0.01 for LPS-treated $Dagla^{-/-}$ group vs vehicle-treated $Dagla^{-/-}$ group; $^{\&\&}$p < 0.01 for LPS-treated $Dagla^{-/-}$ group vs LPS-treated $Dagla^{+/+}$ group. (C) SAG, 2-AG, AA, $PGE_2$, $PGD_2$ and AEA content in brain tissue from $Daglb^{-/-}$ mice and wild-type littermates basally and following exposure to LPS (i.p., 1 mg/kg once per day for 4 days). Data represent average values ± SEM; N = 5-6 per genotype and treatment. *p < 0.05 for vehicle-treated $Daglb^{-/-}$ group or LPS-treated $Daglb^{+/+}$ group vs vehicle-treated $Daglb^{+/+}$ group; $^{\#\#}$p < 0.01 for LPS-treated $Daglb^{-/-}$ group vs vehicle-treated $Daglb^{-/-}$ group. (D) Representative pictures and quantification of microglial activation assessed by Iba-1 staining (a microglia marker that becomes upregulated during inflammatory activation of these cells) in hippocampal regions from $Daglb^{-/-}$ mice and wild-type littermates basally and following exposure to LPS (i.p., 1 mg/kg once per day for 4 days). Scale bar, 50 µm. Data represent average values ± SEM; N = 6 per genotype and treatment. **p < 0.01 for LPS-treated $Daglb^{+/+}$ group vs vehicle-treated $Daglb^{+/+}$ group; $^{\#\#}$p < 0.01 for LPS-treated $Daglb^{-/-}$ group vs vehicle-treated $Daglb^{-/-}$ group; $^{\&\&}$p < 0.01 for LPS-treated $Daglb^{-/-}$ group vs LPS-treated $Daglb^{+/+}$ group. (E) Time course of body temperature changes in $Daglb^{+/+}$ and $Daglb^{-/-}$ mice following LPS (10 mg/kg, i.p.)-induced anapyrexia. Data represent average values ± SEM; N = 6 mice per genotype and treatment. *p < 0.05 for $Daglb^{+/+}$ + Veh vs $Daglb^{+/+}$ + LPS groups; $^{\#}$p < 0.05 for $Daglb^{-/-}$ + Veh vs $Daglb^{-/-}$ + LPS groups; $^{\&}$p < 0.05 for $Daglb^{+/+}$ + LPS vs $Daglb^{-/-}$ + LPS groups.

The following figure supplement is available for figure 6:

**Figure supplement 1.** DAGL inactivation attenuates LPS-induced microglial activation.

---

Previous studies have found that a substantial portion (~40%) of the variance in protein content can be explained by differences in mRNA expression (*Schwanhausser et al., 2011*). Consistent with this conclusion, we found that our activity-based proteomic data displayed good overall correlation with previously reported mRNA expression data obtained by deep RNA-sequencing (*Zhang et al., 2014*). A few notable exceptions of enzymes where transcript and activity were un- or even anti-correlated were, however, identified, and it would be interesting, in future studies, to understand the basis for these differences. One anti-correlated protein – HTRA1 – for instance, is known to be regulated at the level of secretion from cells (*Skorko-Glonek et al., 2013*), and it is possible that individual brain cell types differentially export this enzyme to create discordant measures of mRNA and cellular protein content. Other posttranslational events, such as phosphorylation, could regulate serine hydrolase activity and/or stability. Some of the discrepancies between our chemical proteomic results and previously reported mRNA expression datasets (*Zhang et al., 2014*) may also arise from differences in methodology, given that our studies used primary brain cell cultures instead of acutely isolated neurons and glia. Cells in culture are known to only partially model cells from an in vivo environment, and future ABPP studies using freshly isolated neurons and glia should help to further enrich our understanding of serine hydrolase activities in specific brain cell types.

We were struck by the remarkable number of serine hydrolases that showed differential activity across neurons, astrocytes, and microglia (*Figure 1C*). This observation supports the emerging recognition that neurons and glia are endowed with complementary enzymatic pathways to meet the overall metabolic needs of the nervous system (*Belanger et al., 2011*). Among the most prominently compartmentalized serine hydrolases were those that regulate the eCB 2-AG. This lipid messenger, known to be biosynthesized by two different enzymes—DAGLα and DAGLβ (*Bisogno et al., 2003*)—and with the potential to be degraded by several enzymes—MGLL, ABHD6, ABHD12, and FAAH (*Blankman et al., 2007*, *Marrs et al., 2011*)— is an important regulator of inter-neuronal and neuron-glia interactions (*Navarrete and Araque, 2008*, *2010*, *Martin et al., 2015*, *Viader et al., 2015*). 2-AG signaling also impacts diverse neurophysiological processes, including pain, mood, and neuroinflammation (*Blankman and Cravatt, 2013*, *Murataeva et al., 2014*). Considering the widespread functions of 2-AG, the identification of mechanisms that allow for the selective modulation of

specific aspects of this lipid transmitter's action in the nervous system stands as an important objective. Toward this end, our work establishes that individual cell types in the brain control 2-AG metabolism and signaling by different enzymatic pathways.

We found that microglia, in particular, are equipped with a distinct complement of 2-AG metabolic enzymes – ABHD12 and DAGLβ – the inactivation of which perturbs eCB signaling and crosstalk with eicosanoids specifically in this cell type. ABHD12 plays additional important roles in lipid metabolism in the brain, where the enzyme serves as a major lyso-PS hydrolase (*Blankman et al., 2013*) that is part of an immunomodulatory pathway (*Kamat et al., 2015*) linked to the human neurological disease PHARC (*Fiskerstrand et al., 2010*). It will thus be important, in the future, to dissect in what physiological and disease contexts ABHD12 acts as a 2-AG versus lyso-PS hydrolase. Note also that, while our results indicate a role for ABHD12 in the modulation of microglia endocannabinoid signaling, our experiments do not discriminate the precise identity of the receptors involved. Whether $CB_1R$, $CB_2R$, or even other closely related receptors, mediate the observed ABHD12-regulated 2-AG effect on microglial ERK phosphorylation can be addressed in future studies using lower concentrations of selective $CB_1R$ and $CB_2R$ antagonists or microglia derived from knockout mice lacking these receptors.

DAGLα has been shown to be the principal source for 2-AG production in the nervous system, and disruption of this enzyme impairs many forms of $CB_1R$-dependent synaptic plasticity (*Gao et al., 2010*, *Tanimura et al., 2010*). Our results extend these findings in an important way by demonstrating that DAGLα also regulates neuroinflammatory responses in the brain, in particular, LPS-induced microglial activation. The reductions in LPS-stimulated neuroinflammation observed in *Dagla*$^{-/-}$ mice may reflect, at least in part, crosstalk between eCB and eicosanoid pathways, since basal and LPS-induced prostaglandin synthesis were decreased in these animals. By also identifying DAGLβ as a principal 2-AG synthase in microglia, our work establishes a previously underappreciated and specialized role for this enzyme in 2-AG biosynthesis in the nervous system (*Reisenberg et al., 2012*, *Murataeva et al., 2014*). Our results also do not rule out the presence of DAGLβ-regulated pools of 2-AG in other brain cell types, as might be supported by the significant reductions in this eCB observed in brain tissue from *Dagla*$^{-/-}$ mice treated with MGLL and DAGL inhibitors (*Figure 4F*). Also potentially consistent with a role for DAGLβ in regulating bulk brain 2-AG in the absence of DAGLα, we have been unable to generate viable *Dagla/b* double-knockout mice. Nonetheless, most of the major forms of eCB-dependent synaptic plasticity have been shown to be regulated by DAGLα rather than DAGLβ (*Gao et al., 2010*, *Tanimura et al., 2010*), underscoring the broad role that the former enzyme plays in eCB signaling throughout the nervous system.

That *Daglb* deletion attenuates LPS-induced microglial activation in vivo without altering bulk eCB or eicosanoid content in the brain suggests modulation of restricted pools of 2-AG can impact neuroinflammatory processes while avoiding global effects on eCB signaling. This finding adds to a growing body of work implicating glial proteins and pathways as potential targets for nervous system disease that may have fewer neuron-related adverse side effects (*Barres, 2008*, *Ilieva et al., 2009*, *Milligan and Watkins, 2009*, *Sheridan, 2009*). The remarkable cell type-specific compartmentalization of 2-AG metabolic enzymes, and more broadly of serine hydrolases, may thus prove to be fertile ground for the identification of targets that can safely reverse or slow the course of diverse pathologies of the nervous system.

## Materials and methods

### Materials

FP-rhodamine, FP-biotin, HT-01, KML29, MJN110, KT172, KT195, and DO34 were synthesized in-house as previously described (*Patricelli et al., 2001*, *Chang et al., 2012*, *Hsu et al., 2012*, *Niphakis et al., 2013*, *Ogasawara et al., 2015*). All deuterated lipid standards and substrates were purchased from Cayman Chemicals. Lipopolysaccharide from *E. coli* was purchased from Sigma (0111:B4).

### Primary neuron, astrocyte, and microglia cultures

The primary cell culture protocols used in this study were approved by the Scripps Research Institute Institutional Animal Care and Use Committee (IACUC #09-0041-03). Cortico-hippocampal neurons

were prepared from embryonic day 18 mice from transgenic or wild-type mice as needed. Cortices/hippocampi were dissected, freed of meninges, and dissociated by incubation in Papain/DNase for 20 min at 37°C followed by trituration. Dissociated cortico-hippocampal neurons were then washed with DMEM media supplemented with 10% FBS and 2 mM glutamine, prior to seeding them onto poly-D-lysine coated 10 cm culture dishes in neurobasal medium containing 2% B27 supplement, 2 mM glutamine, and 5 µM 5-fluoro-2′-deoxyuridine at a density of $8 \times 10^6$ cells/dish. A third of the media was exchanged twice per week. Neurons were harvested for proteome isolation after 16 days in vitro in the presence of antimitotics, thus ensuring high neuronal enrichment as confirmed by western blot using neuron-, astrocyte- and microglia-specific markers (Tuj1, GFAP and Iba-1, respectively; data not shown).

Microglia were derived from mixed glial cultures prepared from postnatal day 2-3 mouse forebrains from transgenic or wild-type mice as needed. Briefly, forebrains were dissected, stripped of meninges, and digested in papain/DNAse (20 min at 37°C) followed by 0.25% trypsin (15 min at 37°C) and trituration. Dissociated cells were then cultured for 10 days in poly-D-lysine coated T75 tissue culture flasks in DMEM media supplemented with 10% FBS, 2 mM glutamine, and 5 ng/mL of granulocyte macrophage-colony stimulating factor. After establishment of the astrocyte monolayer, the flasks were shaken for 2 hr at 180 rpm to obtain the loosely attached microglia. Microglia were subsequently plated onto 10 cm dishes at a density of $2\text{-}3 \times 10^6$ cells/dish in Macrophage-SFM media (Gibco) supplemented with 1% FBS and 0.5 ng/mL of granulocyte macrophage-colony stimulating factor. The purity of these microglia cultures was >99% as determined by immunohistochemical quantification of the proportion of Iba-1 positive cells (total cell number determined by DAPI nuclear staining) in six different fields from two separate cultures. Cells were allowed to sit for at least 72 hr prior to harvesting them for proteome isolation.

Following isolation of microglia, established mixed glial cultures were treated with 8 µM cytosine-arabinoside for 3–5 days to kill actively dividing cells (e.g. microglia, fibroblast), and generate an astrocyte monolayer with >85% purity, as determined by immunohistochemical quantification of the proportion of GFAP positive cells (total cell number determined by DAPI nuclear staining) in six different fields from two separate cultures. These astrocytes were subsequently plated onto poly-D-lysine coated 10 cm dishes in DMEM media supplemented with 10% FBS and 2 mM glutamine, and allowed to become confluent. Upon reaching confluence, astrocytes were harvested for subsequent proteome isolation.

## Matings and genotyping of transgenic animals

All animal experiments were carried out in compliance with institutional animal protocols (IACUC #09-0041-03), and mice were housed on a normal 6AM/6PM light/dark phase with ad libitum access to water and food. All mice used in this study were generated by heterozygous matings. *Mgll*$^{-/-}$ (*Viader et al., 2015*), *Abhd12*$^{-/-}$ (*Blankman et al., 2013*), *Dagla*$^{-/-}$ (*Hsu et al., 2012*), and *Daglb*$^{-/-}$ mice (*Hsu et al., 2012*) as well as their wild-type littermates were all in a homogeneous C57Bl/6 background. PCR genotyping of genomic tail DNA was performed using the following primers: *Mgll* 5′- cacctgtctttggagctc ccacc-3′, 5′-cctttctttagggagagtccactgaatgtg-3′, and 5′-ggcagcactgacaaatgtgtctgag-3′ (415-bp product in wild-type mice, 598-bp product in mice with the *Mgll* null allele); *Abhd12* 5′- cagtgctggcctgtcagtcg-3′, 5′-ggtgcccagtgaatggcc-3′, and 5′-taaagcgcatgctccagactgcc-3′ (500-bp product in wild-type mice, 300-bp product in mice with the *Abhd12* null allele); *Dagla* 5′-tgagattggtatcaagacctttg-3′, 5′-ccttgctcctgccgagaaagtatcc-3′, and 5′- gaagaacaggtaaccaggaccat-3′ (300-bp product in wild-type mice, 600-bp product in mice with the *Dagla* null allele); *Daglb* 5′-aaggaggcaaagacagcaaagtgc-3′, 5′-tatcctaggtgcagacagattgtgc-3′, and 5′-aaatggcgttacttaagctagcttgc-3′ (390-bp product in wild-type mice, 195-bp product in mice with the *Daglb* null allele).

## Preparation of mouse tissue and cell proteomes

8-week-old mice were anaesthetized with isofluorane, killed by cervical dislocation, and tissues harvested, immediately flash frozen in liquid nitrogen and kept frozen at −80°C until use. For preparation of proteomes, tissues were homogenized in cold PBS using a bullet blender (Next Advance, Inc.) as per the manufacturer's instructions, or dounced homogenized in an isotonic buffer consisting of 20 mM Hepes, 2 mM DTT, 0.25 M sucrose, and 1 mM $MgCl_2$, pH 7.2 for improved DAGL

visualization. Lysed proteomes were then subjected to a low-speed spin (1400 × $g$, 5 min) to remove debris, and ultracentrifugation (100,000 × $g$, 45 min) to separate membrane and cytosolic fractions. The supernatant was removed and saved as the soluble proteome, while the pellet was washed and resuspended in cold PBS by sonication or in isotonic resuspension solution (20 mM Hepes, 2 mM DTT) by pipetting and saved as the membrane proteome. Total protein concentration for each proteome was determined using a Bio-Rad Dc Protein Assay kit, and proteomes were kept at -80°C until further use. Cultured cell proteomes were prepared in an analogous manner after scraping cells from culture dishes with 1 mL of cold PBS.

## Activity-based protein profiling analysis

50 µL of 1 mg/mL cell or tissue membrane proteome were incubated with the broad-spectrum serine hydrolase probe FP-rhodamine (1 µM final concentration) for 30 min at room temperature or with the DAGL-directed probe HT-01 (1 µM final concentration) for 30 min at 37°C. When necessary, proteomes were pre-treated with inhibitors for 1 hr at 37°C prior to addition of ABPP probes. Probe labeling was terminated by quenching with 4x SDS/PAGE loading buffer, and labeled proteome samples were separated by SDS-PAGE (10% [wt/vol] acrylamide) and visualized by in-gel fluorescence scanning using a flatbed fluorescence scanner (Hitachi FMBio IIe). Gel fluorescence is shown in grayscale, and optical density of the signals was determined using Image-J.

## ABPP-MudPIT sample preparation, MS, and analysis

For the ABPP-MudPIT samples, whole cell proteomes (500 µg/mL in 500 µl of PBS) were labeled with FP-biotin (10 µM) for 2 hr at room temperature. After labeling, the proteomes were precipitated using cold MeOH (2 mL), CHCl3 (0.5 mL) and PBS (1 mL). The resulting cloudy mixture was vortexed and then centrifuged (5000 × g, 15 min, 4°C). The organic and aqueous layers were aspirated leaving a protein disc which had formed between phases. The protein disc was washed with cold 1:1 MeOH:CHCl3 (3 × 1 mL) while intact and then probe sonicated in cold 4:1 MeOH:CHCl3 (2.5 mL). Insoluble proteins were pelleted via centrifugation (5000 × g, 15 min, 4°C) and the supernatant was removed. The remaining pellet was redissolved in 500 µL of 6 M urea in PBS, reduced using Tris(2-carboxyethyl)phosphine (TCEP, 10 mM) for 30 min at 37°C, and then alkylated using iodoacetamide (40 mM) for 30 min at 25°C in the dark. The biotinylated proteins were then enriched by addition of 10% (w/v) SDS (140 µL), followed by 5.5 mL of PBS and 100 µL of PBS pre-washed avidin-agarose beads (100 µL, 1, Sigma-Aldrich) and shaking at 25°C for 1.5 hr. Beads were pelleted by centrifugation (1000 × g, 2 min) and sequentially washed with 0.25% SDS (3 × 10 mL), DPBS (3 × 10 mL) and ddH2O (3 × 10 mL). The beads were transferred to a Protein LoBind tube (Eppendorf) and on-bead digestion was performed using sequence-grade trypsin (2 µg; Promega) in 2 M urea in PBS with 2 mM CaCl$_2$ for 12–14 hr at 37°C (200 µl). Peptides obtained from this procedure were acidified using formic acid (5%) and stored at –80°C before analysis.

MS analysis was performed using a LTQ following previously described protocols (*Jessani et al., 2005*). Peptides were eluted using a five-step multidimensional LC/MS protocol in which increasing concentrations of ammonium acetate are injected followed by a gradient of increasing acetonitrile, as previously described (*Washburn et al., 2001*). For all samples, data were collected in data-dependent acquisition mode over a range from 400–1800 m/z. Each full scan was followed by up to 7 fragmentation events. Dynamic exclusion was enabled (repeat count of 1, exclusion duration of 20 s) for all experiments. The data were searched using the ProLuCID algorithm against a mouse reverse-concatenated, non-redundant (gene-centric) FASTA database that was assembled from the Uniprot database. ProLuCID searches allowed for variable oxidation of methionine (+15.9949 m/z), static modification of cysteine residues (+57.0215 m/z; iodoacetamide alkylation) and accepted only half- or fully-tryptic peptides. The resulting peptide spectral matches were filtered using DTASelect (version 2.0.47), and only half-tryptic or fully tryptic peptides were accepted for identification. Peptides were restricted to a specified false positive rate of <1%. Only proteins detected in at least two out of four replicates for any one cell type preparation, with ≥ 5 spectral counts were used for downstream analysis. Hierarchical clustering analysis shown in *Figure 1C* was performed using CIMminer.

## 2-AG substrate hydrolysis assay

In vitro 2-AG hydrolysis was determined by liquid chromatography-mass spectrometry (LC-MS) monitoring of the generation of AA product. Briefly, 10 µg of primary cell proteomes from $Abhd12^{+/+}$ and $Abhd12^{-/-}$ mice (N = 5 per genotype) were treated with vehicle or 250 nM KML29 prior to incubation with 100 µM 2-AG in PBS (100 µL final volume) for 10 min. Reactions were quenched by addition of 300 µL of 2:1 (vol/vol) CHCl$_3$:MeOH doped with 0.5 nmol of AA-d8 lipid standard, vortexed to mix, and spun at 2000 rpm × 5 min to separate phases. The bottom organic phase was extracted and 20 µL were injected onto an Agilent 6460 triple quadrupole (QQQ) MS. Chromatography was performed on a 50 × 4.60 mm 5-µm Gemini C18 column (Phenomenex) coupled to a guard column (Gemini; C18; 4 × 3.0 mm; Phenomenex SecurityGuard cartridge). The LC method consisted of 0.5 mL/min of 100% buffer A [95:5 (vol/vol) H$_2$O:MeOH plus 0.1% (vol/vol) ammonium hydroxide] for 1.5 min, 0.5 mL/min linear gradient to 100% buffer B [65:35:5 (vol/vol) iPrOH:MeOH:H$_2$O plus 0.1% (vol/vol) ammonium hydroxide] over 5 min, 0.5 mL/min 100% buffer B for 3 min, and equilibration with 0.5 mL/min 100% buffer A for 1 min (10.5 min total run time). MS analysis was performed in negative scanning mode with an electrospray ionization (ESI) source using the precursor to product ion transition and collision energies for AA (303, 303, 0, negative) and AA-d8 (311, 311, 0, negative). Dwell times for each lipid were set to 100 ms, and the following MS parameters were used: capillary voltage = 3.5 kV, drying gas temperature = 350°C, drying gas flow rate = 9 L/min, and nebulizer pressure = 50 psi, sheath gas temperature = 375°C, and sheath gas flow rate = 12 L/min. AA release was quantified by measuring the area under the peak in comparison with the AA-d$_8$ internal standard and correcting for non-enzymatically formed AA present in heat inactivated (10 min at 90°C) control reactions, and relative 2-AG hydrolytic activity for conditional $Abhd12^{-/-}$ proteomes was calculated by comparing to activity of wild-type proteomes.

## Lipid measurements

Metabolite levels were quantified by multiple reaction monitoring (MRM) of each lipid species using and Agilent 6460 QQQ instrument. For cultured cells, lipids were extracted from each cell type by scraping cells from dish in 1.1 mL of cold PBS, of which 100 µL were saved to measure protein concentration for normalization, prior to re-scraping with 1 mL of cold MeOH. The 2 mL of cell lysate were then transferred to 2 mL of CHCl$_3$ with 20 µL of formic acid and the following lipid standards: 1 nmol of 2-AG-d$_5$, 0.5 nmol of AEA-d$_4$, 1 nmol of AA-d$_8$, and 0.5 nmol of PGE2-d$_9$. After centrifugation at 2000 rpm for 5 min, the organic bottom fraction was carefully collected and dried under a nitrogen stream. Lipids were resolubilized in 140 µL of 1:1 (v/v) CHCl$_3$:MeOH, 20 µL of which were injected for mass spectrometry analysis. When necessary, 5 mL of media were also extracted in an analogous manner by combining with 5 mL of MeOH and 10 mL of CHCl$_3$. For LPS studies, lipids were similarly extracted from 2-3 × 10$^6$ microglia pre-treated with inhibitors as needed for 3 hr prior to addition of 100 ng/mL of LPS for 4 hr.

For brains, 2 month-old $Dagla^{-/-}$, $Daglb^{-/-}$, or corresponding wild-type littermates were injected with LPS in PBS (1 mg/kg, once a day for 4 days, i.p.) or with PBS vehicle alone. On day 5, mice were sacrificed by cervical dislocation, and their brains rapidly removed and immediately flash frozen in liquid nitrogen. One brain hemisphere was then weighted and dounce homogenized in 8 ml of cold 2:1:1 (v/v/v) CHCl$_3$:MeOH:PBS doped with the following lipid standards: 1 nmol of 2-AG-d$_5$, 0.5 nmol of AEA-d$_4$, 1 nmol of AA-d$_8$, 1 nmol of SAG-d$_8$, and 0.5 nmol of PGE2-d$_9$. Note that brains were not allowed to thaw prior to contact with organic solvents. Brain homogenates were vortexed and centrifuged at 2000 rpm for 5 min. The organic bottom fraction was carefully collected, and the remaining solution was re-extracted by adding another 2 mL of CHCl$_3$ with 20 µL of formic. The organic fractions from both extractions were combined, and dried under a nitrogen stream. Lipids were resolubilized in 200 µL of 1:1 (v/v) CHCl$_3$:MeOH, 10 µL of which were injected for mass spectrometry analysis.

LC separation of lipid metabolites was performed on a 50 × 4.60 mm 5 µm Gemini C18 column (Phenomenex) coupled to a guard column (Gemini; C18; 4 × 3.0 mm; Phenomenex SecurityGuard cartridge). Mobile phase A consisted of 95:5 (vol/vol) H$_2$O:MeOH and mobile phase B consisted of 60:35:5 (vol/vol) iPrOH:MeOH:H$_2$O, with 0.1% formic acid or ammonium hydroxide added to both mobile phases to assist in ion formation in positive and negative ionization modes, respectively. For the targeted detection of MAGs, NAEs, and DAGs in positive mode, each run started at 0.1 mL/min

of 100% A. At 5 min, the solvent was immediately changed to 60% B with a flow rate of 0.4 mL/min and increased linearly to 100% B over 15 min. 100% B was allowed to flow at 0.5 mL/min for an additional 8 min prior to equilibrating for 3 min with 100% A at 0.5 mL/min. For targeted detection of eicosanoids and fatty acids in negative mode, each run started at 0.1 mL/min of 100% A. At 3 min, the flow rate was increased to 0.4 mL/min with a linear increase of solvent B to 100% over 17 min. 100% B was allowed to flow at 0.5 mL/min for 7 min prior to equilibrating for 3 min with 100% A at 0.5 mL/min. MS analysis was performed in either positive or negative scanning mode with an electrospray ionization (ESI) source, and the following parameters were used to measure the indicated metabolites by SRM (precursor ion, product ion, collision energy in V, polarity): 2-AG (379, 287, 8, positive); 2-AG-d5 (384, 287, 8, positive); AEA (348, 62, 11, positive); AEA-d4 (352, 66, 11, positive); SAG (662.5, 341, 40, positive); SAG-d8 (670.5, 341, 40, positive); AA (303, 303, 0, negative); AA-d8 (311, 311, 0, negative); $PGE_2/D_2$ (351, 271, 4, negative); $PGE_2/D_2$-d9 (360.5, 280, 4, negative). Dwell times for each lipid were set to 100 ms, and the following MS parameters were used: capillary voltage = 3.5 kV, drying gas temperature = 350°C, drying gas flow rate = 9 L/min, and nebulizer pressure= 50 psi, sheath gas temperature = 375°C, and sheath gas flow rate = 12 L/min. Metabolite species were quantified by measuring the area under the peak in comparison with the appropriate unnatural internal standard and normalizing for wet tissue weight or protein concentration for cultured cells.

## Cytokine analysis

Cytokines were measured using a DuoSet ELISA kit (R&D systems) as per the manufacturer's instructions. For cultured cells, 100 μL of media were used.

## Western blotting

For western blotting, samples (30 μg protein) were separated by SDS-PAGE [10% (wt/vol) acrylamide] then transferred to a nitrocellulose membrane. The membrane was then blocked in 5% milk in 0.5% TBS-Tween and incubated overnight with P-Erk1/2 (1:1000, rabbit, Cell Signaling) and Erk 1/2 (1:1000, rabbit, cell signaling) primary antibodies. Following incubation with IRdye680 secondary antibody (1:5000, Licor Biosciences), membranes were visualized with a Licor Odyssey CLx near-infrared imager.

## Immunohistochemistry

Mice were deeply anesthetized using isofluorane and perfused with PBS followed by 4% (wt/vol) paraformaldehyde. Brains were then carefully dissected, postfixed in 4% paraformaldehyde overnight, cryoprotected in 30% (wt/vol) sucrose, and rapidly frozen on dry ice. Free floating coronal sections 40 μm in thickness were cut on a Leica CM1850 cryostat. Frozen, free-floating sections were blocked with Bloxall (Vector labs, 10 min) and 0.2% Triton-X100 and 3% goat serum in PBS (blocking solution, 1 hr) prior to O/N incubation at 4°C with a rabbit anti-ionized calcium-binding adaptor molecule (Iba)1 primary antibody (1:500 dilution in blocking solution; Wako). Sections were then incubated with secondary antibody (anti-rabbit biotin; Vector Laboratories; 1:300 dilution of 1.5 mg/mL stock) in 0.5% (wt/vol) BSA in 0.1 M PB for 1 hr at room temperature and developed with ABC Elite Vectastain (Vector Laboratories). After staining, sections were mounted in Permount and imaged using a Leica SCN400 whole slide scanner. Images were processed using ImageJ and Photoshop using global adjustments in brightness and contrast. Iba1-positive microglia were quantified in matching sections from at least 5 mice per genotype using ImageJ by first converting images to be counted to 8 bit greyscale and applying global adjustments in brightness and contrast in the same way to the whole set of images being analyzed. The area for all cells being quantified was then highlighted using ImageJ's Threshold macro, setting the same range of pixel intensities for the entire set of images being processed. Total microglia area in the resulting binary image was determined using ImageJ's Particle Analyzer macro. Area for all particles larger than 10 pixels was included in the quantification.

## Anapyrexia and core body temperature measurements

Core body temperature was measured by radiotelemetry as previously described (*Sanchez-Alavez et al., 2015*). Briefly, mice were anesthetized with isofluorane (induction 3–5%, maintenance

1–1.5%) and surgically implanted with radiotelemetry devices (TA-F20, Data Sciences, St. Paul, MN) into the peritoneal cavity for core body temperature (CBT) and locomotor activity (LA) evaluation. Following surgical implantation and appropriate wound closure, the animals were allowed to recover for 2 weeks and then subjected to telemetry recordings. Mice were individually housed in a Plexiglas cage in a room maintained at $25 \pm 0.5°C$. The cages were positioned onto the receiver plates (RPC-1; Data Sciences) and radio signal reporting CBT and LA information from the implanted transmitter was recorded continuously with a fully automated data acquisition system (Dataquest ART, Data Sciences, St. Paul, MN). Access to food and water was ad libitum and the light:dark cycle was of 12 hr:12 hr. Anapyrexia was induced by intraperitoneal injection of *Bacterial lipopolysaccharides (LPS)* (0127:B8, Sigma, St. Louis, MO) at a dose of 10 mg/kg in saline.

## Data analysis and statistics

Data are shown as the mean ± SEM. A Student's t test (unpaired, two-tailed) was used to determine differences between two groups. Data that included more than two groups were analyzed by one-way ANOVA, with post hoc Sidak's multiple comparisons test. Correlations were determined by two-tailed Perason's correlation. All statistical analyses were conducted using Excel or GraphPad Prism version 6, and a p value <0.05 was considered significant throughout. No statistical methods were used to predetermine sample sizes, which were comparable to those in previous publications (*Viader et al., 2015*) and based on previous knowledge of the variability associated with the different experiments and the expected differences.

## Acknowledgements

We thank Dr. Ken Hsu (U Virginia) and members of the Cravatt lab for helpful discussions and technical assistance. This work was supported by the National Institutes of Health grants DA033760 (BFC), and GM109315 (AV). The authors declare no competing financial interests.

## Additional information

### Competing interests

BFC: Reviewing editor, *eLife*. The other authors declares that no competing interests exist.

### Funding

| Funder | Grant reference number | Author |
| --- | --- | --- |
| National Institute on Drug Abuse | DA033760 | Benjamin F Cravatt |
| National Institute of General Medical Sciences | GM109315 | Andreu Viader |

The funders had no role in study design, data collection and interpretation, or the decision to submit the work for publication.

### Author contributions

AV, Conception and design, Acquisition of data, Analysis and interpretation of data, Drafting or revising the article; DO, CMJ, MSA, SM, WN, Acquisition of data, Analysis and interpretation of data; BC, Analysis and interpretation of data, Drafting or revising the article; BFC, Conception and design, Analysis and interpretation of data, Drafting or revising the article

### Ethics

Animal experimentation: The primary cell culture protocols and animal experiments in this study were approved by the Scripps Research Institute Institutional Animal Care and Use Committee (IACUC #09-0041-03).

## Additional files

**Supplementary files**
• Supplemental file 1. Complete proteomic data for ABPP-MudPIT experiments of primary cultured neurons, astrocytes, and microglia using the serine hydrolase-directed activity-based probe FP-biotin.

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
