## [Decision Letter]

Thank you for submitting your work entitled "A chemoproteomic atlas of brain serine hydrolases identifies cell type-specific pathways regulating neuroinflammation" for consideration by *eLife*. Your article has been reviewed by two peer reviewers, and the evaluation has been overseen by a Reviewing Editor and Aviv Regev as the Senior Editor. One of the two reviewers, Nephi Stella, has agreed to reveal his identity.

The reviewers have discussed the reviews with one another and the Reviewing Editor has drafted this decision to help you prepare a revised submission.

Summary:

In this interesting study, Viader et al. assess the distribution of serine hydrolases in cultured mouse neurons, astrocytes and microglia. They use activity-based protein profiling combined with multidimensional LC-MS and find that distinct groups of serine hydrolases are enriched in specific neural cell types. Comparison of serine hydrolase expression profiles in neural cell types correlates well with the recently reported RNA-seq database from the Barres lab (Zheng et al., 2014), with some noticeable exceptions.

Of particular interest are results showing the clear compartmentalization in the functionality of the endocannabinoid/eicosanoid signaling between neurons and microglia. Furthermore, the in vivo testing of these particular signaling pathways using selective inhibitor and knockout technology provides a convincing validation.

However, there are a few textual changes that will need to be made before the manuscript is ready for acceptance.

Essential revisions:

1) With regards your cell culture preparations, please provide more specifics and the actual data of their characterization. What were the readouts (specific antibodies) and the sampling approach used to determine cell numbers? What non-neuronal cells?

2) Microglial cells were cultured in the presence GMCSF, which induces a dendritic-like phenotype in these cells. Please comment on how this influences the interpretation of these results.

3) Add to the Discussion something on the limitation of this study. Specifically, the initial profiling was performed on cells in culture, which are known to only partially model in situ CNS cells. Several key validation steps of these results will have to be performed in future studies, including confirming key cell culture results using freshly isolated CNS cells, for example using FACS. Further studies using cell specific genetic alteration will also provide further validation of the results.

4) Regarding IHC analysis, please provide methodological specifics: were masks generated to outline each cell or the number of positive pixels counted? How were the background signal and global adjustments set?

5) In Figure 2, involvement of CB1 and CB2 receptors is tested with antagonists applied at 5 µM, which is 2 orders of magnitude higher than their respective potency at these target (and thus too high to leverage their selectivity). Testing the response obtained with lower concentrations of these antagonists would be more convincing and informative.

---

## [Author Response]

Essential revisions:

1) With regards your cell culture preparations, please provide more specifics and the actual data of their characterization. What were the readouts (specific antibodies) and the sampling approach used to determine cell numbers? What non-neuronal cells?

Our glial culture preparations were characterized by immunocytochemistry and western blotting. Iba-1 and GFAP were used as markers for microglia and astrocytes, respectively. As shown in Figure 7 and noted in the Methods sections, our microglia and astrocyte cultures are routinely ~99% and ~85% pure respectively, as determined by quantifying the proportion of Iba-1 or GFAP positive cells counted in six different fields from two separate cultures (total cell number determined by DAPI nuclear staining and/or actin staining).

Our neuron cultures are grown in the presence of anti-mitotics (5 μM 5-fluoro-2′-deoxyuridine) for over 2 weeks, thus also ensuring high neuronal enrichment. This was confirmed in the western blot below, which shows that cell lysates from our neuronal cultures strongly express the neuronal marker Tuj1 but lack astrocytic or microglial markers (GFAP and Iba-1). A brief description of the characterization of our cell culture preparations has now been included in the Methods section.

Author response image 1.**DOI:**
http://dx.doi.org/10.7554/eLife.12345.015

2) Microglial cells were cultured in the presence GMCSF, which induces a dendritic-like phenotype in these cells. Please comment on how this influences the interpretation of these results.

We recognize the reviewers’ concern regarding the potential effect of culturing microglia in the presence of GM-CSF, since this growth factor is known to induce a dendritic-like phenotype in these cells. The large number of microglia cells required for our proteomic and metabolomic studies (we often needed >60 million cells), however, made the use of GM-CSF necessary to increase the microglia cell yield and accomplish our experimental goals. Given that cultured cells in general and GM-CSF-exposed microglia in particular only partially model CNS cells in vivo, the results from the present study need to be considered as a starting point to identify potentially interesting serine hydrolase activities in different brain cell types. The functional relevance of individual enzymes, however, and as we showed in this study, requires additional experimental validation. Note that some of the limitations related to using cultured microglia are now described in the Discussion section of the revised manuscript.

3) Add to the Discussion something on the limitation of this study. Specifically, the initial profiling was performed on cells in culture, which are known to only partially model in situ CNS cells. Several key validation steps of these results will have to be performed in future studies, including confirming key cell culture results using freshly isolated CNS cells, for example using FACS. Further studies using cell specific genetic alteration will also provide further validation of the results.

We agree with the reviewers that the present work could be strengthened by generating our chemoproteomic profiles using freshly isolated CNS cells. As the sensitivity of our chemoproteomic methods improves and the yields of acute brain cell type isolation increase, such studies should become technically feasible. We have now included the following sentence in the Discussion to acknowledge some of the limitations associated with the use of brain cells in culture for our proteomic profiles: “Note that some of the discrepancies between our chemical proteomic results and previously reported mRNA expression datasets (Zhang et al., 2014) may also arise from differences in methodology, given that our studies used primary brain cell cultures instead of acutely isolated neurons and glia. Cells in culture are known to only partially model cells from an in vivo environment, and future ABPP studies using freshly isolated neurons and glia should help to further enrich our understanding of serine hydrolase activities in specific brain cell types.”

4) Regarding IHC analysis, please provide methodological specifics: were masks generated to outline each cell or the number of positive pixels counted? How were the background signal and global adjustments set?

The Methods section now includes a description of the IHC analysis. Briefly, images to be counted were converted to 8 bit greyscale and global adjustments in brightness and contrast applied in the same way to the whole set of images being analyzed. The area for all cells being quantified was then highlighted using ImageJ’s Threshold macro, setting the same range of pixel intensities for the entire set of images being processed. Total microglia area in the resulting binary image was determined using ImageJ’s Particle Analyzer macro. Area for all particles larger than 10 pixels was included in the quantification.

5) In Figure 2, involvement of CB1 and CB2 receptors is tested with antagonists applied at 5 µM, which is 2 orders of magnitude higher than their respective potency at these target (and thus too high to leverage their selectivity). Testing the response obtained with lower concentrations of these antagonists would be more convincing and informative.

The reviewers are correct to point out that at 5 μM AM630 and rimonabant may not be completely selective for CB1 and CB2 receptors. We chose a somewhat high concentration of AM630 and rimonabant for this set of experiments to ensure complete blockade of any cannabinoid receptors potentially involved in microglial 2-AG signaling. Note also that both AM630 and rimonabant are often used in the low micromolar range in the cannabinoid research literature, thus while 5 μM is a high concentration for these inhibitors, it is not an unreasonable one. The following sentence has been added to the Discussion to address the reviewers’ concern: “Note also that, while our results indicate a role for ABHD12 in the modulation of microglia endocannabinoid signaling, our experiments do not discriminate the precise identity of the receptors involved. Whether CB_1_R, CB_2_R, or even other closely related receptors, control the observed ABHD12-regulated 2-AG effect on microglial ERK phosphorylation can be addressed in future studies using lower concentrations of selective CB_1_R and CB_2_R antagonists or microglia derived from knockout mice lacking these receptors.”